# Lean Workbook: A large-scale Lean problem set formalized from natural language math problems

**Huaiyuan Ying**[12*]**, Zijian Wu**[13*]**, Yihan Geng**[14*]**, Jiayu Wang**[1]**, Dahua Lin**[1]**, Kai Chen**[1]

[1]Shanghai AI Laboratory, [2]Tsinghua University, [3]Shanghai Jiao Tong University, [4]Peking University
`internlm@pjlab.org.cn`

## Abstract

Large language models have demonstrated impressive capabilities across various natural language processing tasks, especially in solving mathematical problems. However, large language models are not good at math theorem proving using formal languages like Lean. A significant challenge in this area is the scarcity of training data available in these formal languages. To address this issue, we propose a novel pipeline that iteratively generates and filters synthetic data to translate natural language mathematical problems into Lean 4 statements, and vice versa. Our results indicate that the synthetic data pipeline can provide useful training data and improve the performance of LLMs in translating and understanding complex mathematical problems and proofs. Our final dataset contains about 57K formal-informal question pairs along with searched proof from the math contest forum and 21 new IMO questions. We open-source our code at `https://github.com/InternLM/InternLM-Math` and our data at `https://huggingface.co/datasets/InternLM/Lean-Workbook`.

## 1 Introduction

*I do believe that problems are the heart of mathematics. – P. R. Halmos*

Proving theorems is one of the most fundamental goals in mathematics, which requires complex math reasoning and a rich store of math knowledge. Recently, large language models (LLMs) [15, 27, 21, 3, 6, 25, 36] have made great progress in solving grade-school [5] and even high-school level math problems [8] through chain-of-thought reasoning [30]. LLMs can also interact with proof assistants including Lean [19], Coq [28], or Isabelle [22] to prove theorems. However, the performance of theorem proving is not satisfying with LLMs [37].

One reason for this weakness is data sparsity. The mainstream approach for LLMs in learning theorem proving is through expert iteration[1, 32, 14, 23, 33]. LLMs search the proof in the given math problem and statement set like MiniF2F [37] and Mathlib [18] and learn from their success trajectories. However, the amount of data in MiniF2F is limited because formalizing problems requires significant labor from human experts. Though Mathlib is a very large dataset that contains the formalization of different math subjects in Lean, it mainly proves fundamental math theorems instead of contest-level problems. Therefore, an initial step toward a better automatic theorem-proving model is to create enough high-quality formalized statements.

In this work, we present Lean Workbook: an iterative autoformalization pipeline, together with a large-scale Lean problem set. We train our autoformalization model based on active learning. At each turn, we use our model to translate natural language problems into formal statements and back-translate to natural language problems collected from the math contest forum[2]. We use Lean

---

[*]Work done during internships at Shanghai AI Laboratory.
[2]`https://artofproblemsolving.com/community`

38th Conference on Neural Information Processing Systems (NeurIPS 2024) Track on Datasets and Benchmarks.

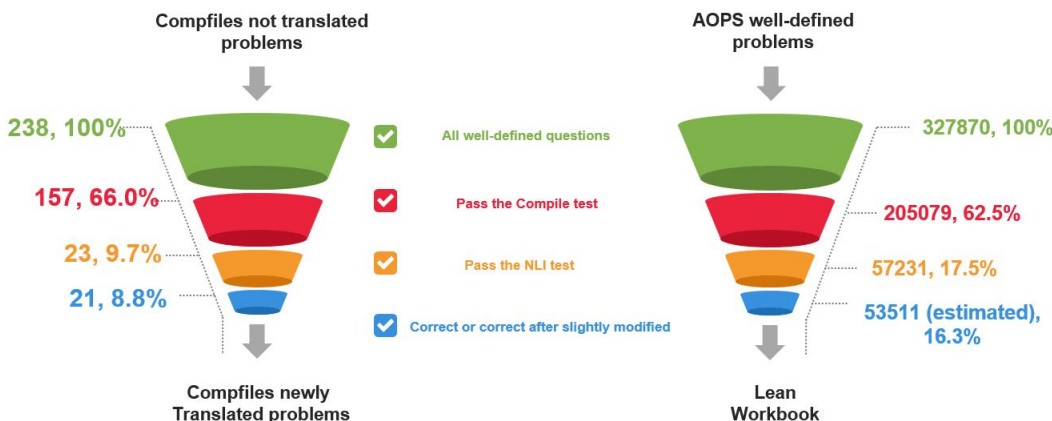

Figure 1: The data contribution of our Lean Workbook pipeline. Three rounds of filtering will mostly ensure the accuracy of output data. By applying the pipeline to the AOPS and the Compfiles data sources respectively, we derive 21 formalized IMO questions and about 57k synthetic training data for autoformalizaion.

compiler and Natural Language Inference (NLI) to check if it is a valid formalization. We sample invalid formalization and require human experts to modify them into a valid formalization and add them to the training set. Through the supplement of human-labeled data pairs, the translation model gradually learned to translate between Lean 4 formal language and natural language questions of different types of problems. We autoformalized 57K math problems in the final round. Manual examination reports an accuracy of 93.5% of a random sample of the Lean Workbook. The same filtering process produces 21 new formal statements of the IMO questions which do not appear in Compfiles [3].

In conclusion, our contribution can be summarized as follows:

- We propose an active learning pipeline for autoformalizing natural language questions.
- We open-source our translation model and pipeline, which can be used for autoformalizing diverse topics of math statements.
- We open-source a dataset containing 57k formalized math problems (5k of them have formal solutions) which can be used for autoformalization and auto theorem proving.
- We formalize 21 new IMO questions that have not appeared in Compfiles.

## 2 Preliminaries

Formal proof involves establishing claims that are expressed in precise mathematical terms in programming languages. Lean 4, which is the latest version of Lean Theorem Prover, aims to provide an open-source platform for correct and maintainable code for formal verification. In the Lean language, users can define a theorem and prove it by tactics or pretend to complete the proof using "sorry". Lean 4 will return a "No goals" signal if the proof is completed.

The Mathlib is a user-maintained library for Lean 4. With the help of Mathlib, we can utilize other's previously formalized theorem or function to state our theorem and proof process. Therefore, in the following paragraphs, we default talk about applying Mathlib as MiniF2F does in its environments.

Our work focuses on the translation of questions instead of proof. Therefore, we will always use "sorry" for the proof. A typical Lean 4 statement looks as follows. The 'theorem' declares a type of this proposition, followed by the theorem name. Then it specifies all the variables and their types, along with several conditions separated by brackets. Finally, the conclusion starts after the colon, and ":= by sorry" finishes the proof. Here is an example:

---

[3]`https://github.com/dwrensha/compfiles`

```
theorem ex_1 (n p : ℕ) (hp: Nat.Prime p) (h₁ : p | n) : { (x, y) : ℕ × ℕ | x + y =
    n ∧ Nat.gcd x y = p }.Finite := by sorry
```

# 3 Related works

## 3.1 Autoformalization

Autoformalization [32] refers to translating natural language math statements or proofs into formal languages. Previous works have autoformalized different levels of mathematics including grade-school level [9, 20], high-school contest level [32, 16], and undergraduate level mathematics [2, 11] utilizing in-context learning or fine-tuned LLMs. Our works focus on formalizing high-school contest-level math problems with a much larger scale. A similar and concurrent work is DeepSeek-prover [33] which translates a large-scale Lean problem set from high-school problems. Compared to DeepSeek-prover, we apply active learning to reduce incorrect formalization and we manually evaluate our proposed dataset and find a high formalization accuracy.

## 3.2 Automatic Theorem Proving

Using large language models to automatically prove math theorems do not have a unified approach. The mainstream approach is to conduct a best-first search or tree search on proof states [7, 24, 14, 13, 35, 31, 3, 36]. This approach can prevent to generate invalid tactics since they will be rejected by the compiler immediately, but the model cannot predict tactics based on an overall perspective. In contrast, another approach is to leverage LLMs to generate the whole proof based on itself[33] or human's proof[12, 29].

## 3.3 Data curation

In the field of formal language, several works have established their methods of development and curation of datasets. Runtime information has proved to be beneficial for the construction of datasets containing whole proofs, which can be extracted via tools such as LeanDojo and CoqGym[35, 34]. Meanwhile, human annotations and human interaction with language models can provide valuable assistance towards stepwise proofs[26, 31]. Aside from large-scale datasets, there are also highly accurate benchmarks for formally verifying proofs[17, 37]. Our work shares a similarity with these works in that human experts and LLMs contribute together to the autoformalization data construction.

# 4 Data construction pipeline

In this section, we will detailedly describe the whole pipeline for iteratively translating and filtering correct samples as in Figure 4, and then demonstrate the final dataset construction procedure.

## 4.1 First-round pipeline

We first collect Lean 4 formal statements with their corresponding natural language questions from MiniF2F[37][4] and ProofNet[2][5]. Since we do not test autoformalization on MiniF2F and ProofNet, we use all samples from these two datasets.

The proof will be declared using ":= sorry". All the sample pairs would be organized from two directions into the training data to achieve a two-way translation between the formal language and natural language. We also include multi-task Lean 4 instruction data including proving theorems, predicting the next tactics, and explaining the Lean proof using natural languages like [7, 11] during training.

The training data can be split into proof questions and questions with an exact gold answer. However, Lean 4 only supports proof questions, so we rephrase all the solution questions by adding a proof goal. Concretely, we append "Show that it is {answer}." to the original natural questions, while the proof goal in formal statements is changed to prove the solved answer should be the gold one.

---

[4]We use the version of `https://github.com/rah4927/lean-dojo-mew`. Under Apache Licence.
[5]We use the version of `https://github.com/rahul3613/ProofNet-lean4` Under MIT Licence.

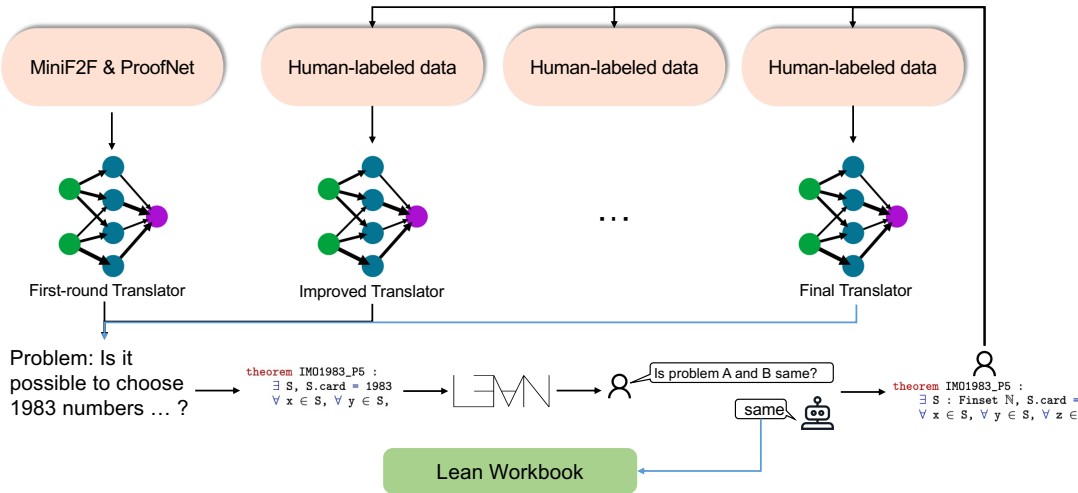

Figure 2: The main flowchart of our pipeline. Starting from the initial training data, we finetune our translation model which is then applied to a natural language problem set. The translated data is filtered by Lean 4 compiling, backtranslate and NLI test, and human diagnostic. We manually conclude patterns and accordingly add training data into the model fine-tuning in the next iteration. The filtered samples are exported if the labelers consider them to reach enough accuracy.

The first-round data collection is fed to our translation model, which is initialized from InternLM-Math-Plus-20B [10] which has been pre-trained on Lean-related datasets. The model is fine-tuned for three epochs with a learning rate of $4e - 5$, with two different but fixed prompts for each translation direction. This translated data serves as a starting point for further iteration. Fine-tuning uses 32 A100 GPUs and can be finished within several hours.

After training a translation model, we want to improve our model on formalizing problems with diverse math topics. We collect math problems from the math contest forum [6] as our active learning dataset. It contains problems from middle to high school math, with varying difficulties up to Olympiad levels. We utilize Qwen-1.5-14B-Chat [4] to extract the question, solution, and gold answer from each post of the forum with the following prompt.

*You are a data labeler. Here is a discussion between math students. It may contain several problems and several solutions. Please extract them in a JSON format. Each problem is an element and has keys including problem (str, you should not miss any assumption like non-negativity of numbers, be formal), answer (return numbers as a string for calculation problems and return an empty string for proof problems), and tags (list of str). Tags should identify the category of this math problem. Possible tags contain: equation, inequality, number_theory, algebra, probability, combination, trigonometry, and etc.*

It is observed that some kinds of problems are not suitable for formalizing. Meanwhile, the extraction process turns out to be unstable and gives badly-stated problems. Firstly, we only keep those with one of the following tags: inequality, number theory, trigonometry, modular arithmetic, induction, functional equation, complex numbers, and polynomial. Secondly, we query the Qwen model whether the problem is ill-defined with the following prompt.

*Please check whether the following math problem is well-defined? Please follow the rules: 1. Consider each condition given in the problem, it is not well-defined one variable is used without definition anywhere in the question.*
*2.The problem is not well-defined if it contains more than one goal or no clear goals to solve.*
*3. Note that inequalities may omit the statement that $x, y, z, a, b, c$ are real numbers, but they are well-defined, do not judge them to be ill-defined.*
*4. Please reply \*\*well-defined\*\* or \*\*ill-defined\*\* in the final sentence with bold format, be sure not to fail well-defined questions.*

---

[6]https://artofproblemsolving.com/community

We filter out the ill-defined questions. The manual revision shows almost no ill-defined questions are left, though a small part of well-defined ones are wrongly omitted. After such cleaning, we use our initial translation model to translate all filtered problems into formal statements.

The well-defined subset contains 6652 different tags in total, with 223 tags containing over 100 samples. These tags cover a large range of questions from contest-level knowledge points to high-school courses. More than three-fourths of the samples are labeled with algebra-relevant tags, while geometry-related tags are rarely witnessed. It is also noticed that some tags are wrong, especially the "number theory" tag is often allocated to inequality problems. Following these findings, we will remain keeping working with tags over 100 samples in later analysis and will pay special attention to wrong tags during manual diagnostic.

## 4.2   Data Diagnostic and Iteration pipeline

**Compiling Correctness test** To ensure the accuracy of the formal statements produced by our translation pipeline, each translated formal theorem undergoes a correctness check within a Lean 4 environment. Initially, the theorem statements are verified independently, using a placeholder "by sorry" for the proofs, to filter out incorrect statements in advance. The complete theorem, including proofs, is then examined. The major bottleneck of this step is the compiling cost of Lean 4 projects. To facilitate the process, we build up a Lean 4 read-eval-print loop (REPL), utilizing Lean 4's runtime meta-programming facility, which allows for the verification of Lean 4 statements in an interpreted mode. The correctness test program can be executed in a multi-process style and can be finished in one hour with a 32-core CPU. Our test environment is based on Lean v4.8.0-rc1 with Mathlib4 of the same version (which can be cloned by specifying the tag v4.8.0-rc1).

**Data Filtering** Firstly, the synthetic translation from all problems is processed by the compiling correctness test. However, it is usually witnessed that a correctly compiled translation actually does not follow the original question. The second step of filtering is based on the back translation ability of our model. After the formal statement is translated back into natural questions, we can turn to using a general domain LLM to leverage its Natural Language Inference ability. In our pipeline, we still query the Qwen-1.5-14B-Chat to judge if the original question is the same as the back-translated version. If we do not get a positive response, the sample is marked as needing human revision and correction. The prompt writes as:

*Please check following two math problems is same or different? Please consider each statement in two problems, they are different if any statement is different. Please point out any differences you found. Please reply \*\*same\*\* or \*\*different\*\* in the final sentence with bold format.*

**Diagnostic and Human labeling** Diagnostic for the data mainly focuses on two kinds of samples: the ones that do not pass the compiling correctness test and the ones that pass the test but do not prove to be a correct translation with a positive NLI feedback. The other samples that pass the NLI test are considered to be correct for now. In the first three rounds of our iterations, these two kinds of samples both have relatively obvious patterns. Thus we conclude and modify them accordingly with three human experts who are familiar with both Lean and contest-level math problems [7]. Each evaluator was assigned an equal number of problems, ensuring a balanced distribution of the workload. On average, each problem required approximately two to five minutes for evaluation.

The manually modified samples are added to the training data, and a new translation model is fine-tuned for the next round of generating and filtering synthetic samples for human diagnostics. These two processes are the same as in the previous paragraph. Each iteration will add an average of about 30 human-labeled samples into the training data, addressing the current model's weakness.

After several rounds, it becomes difficult to conclude patterns. We change our diagnostic mode and randomly sample math problems by tags. By manually checking the samples, we will add the correct or modified ones into the training data, and record the correct rate in the samples which pass the NLI test. Each iteration will gain more samples passing the NLI test and an increase in the correct rate. We stop our iteration after six rounds when the correct rate in sampled data almost reaches 95%, and we add 341 problems into the training set during iterations.

---

[7]They all won a prize in the National Mathematical Olympiad Contest.

# 5 Results

This section will introduce our evaluation metric, dataset statistics, and case studies together with our analysis of the cases.

## 5.1 Evaluation setting

Unlike auto theorem proving which depends totally on Lean 4 programming to check the accuracy, our evaluation for both the pipeline and the final translation datasets includes the three metrics: (1) Compile pass number (CPN): The number of all generated formal statements that can be correctly complied using Lean 4 under the environment of Mathlib. (2) NLI pass number (NPN): The number of generated formal statements that simultaneously can be compiled and the back translation is considered the same as the original questions by the model performing the NLI task. (3) Correct translation rate: The proportion of generated formal statements which is considered by human experts as a precise translation in those passing the NLI test. In real-world settings, it is too consuming to manually review all the synthetic data, so the reported value is the rate on a sampled subset based on question types.

## 5.2 Dataset Statistics and Evaluation Results

The original active learning dataset has 1088678 questions, among which 458692 questions are considered well-defined. The ill-defined questions come from an incomplete extraction from the website, or a post containing attempts and parts of solutions for a specific problem. After filtering the tags, 327870 questions are selected to be formalized in our experiments.

After six rounds of iteration, our model outputs 205079 questions that pass the compiling correctness test, among which 57231 translations pass the NLI test. We randomly select five to ten samples for each common tag (tag with over 100 samples), and manually check whether they are truly correct. The results are in Table 1. For the most common three tags, we sample 10 questions and all of them achieve a sampled accuracy over 90%. The other tags each stand for a special kind of problem showing up in mathematical contests and college examinations, among which almost all tags have at most one wrong translation.

We use InternLM-Math-Plus to search proofs in the Lean Workbook by sampling multiple whole proofs and checking by our correctness checker. We sample 1024 proofs for each problem and we solve 4898 of them (i.e. the Pass@1024 is 8.6%) which is significantly harder than MiniF2F. We will also open-source these solutions to help improve automatic theorem proving.

Though the overall accuracy has reached a high level, some kinds of mistakes still occasionally happen, which is also indicated in the table as not all the tags have 100% accuracy. On the other hand, a number of patterns have been corrected during iterations. These patterns contain compiling errors like conflict type and functions and continued inequalities, whose correction can significantly increase the CPN. Our model demonstrates good learning ability in these samples. If the model does not know how to translate a kind of problem, three manually written statements would help the model learn how to translate it. However, when it comes to the errors of interpreting a contest problem, the effectiveness of iteratively adding human-labeled samples decreases. For example, when one integer is divided by another integer without specifying the type, the Lean language will return the floor of the true quotient. So we add the type : $\mathbb{R}$ to them, but the model can only perform correctly about half the times. This may be attributed to the confusion from real number divisions but written in the same form. Other instances include minimal/maximal problems where the model only states one-side inequality but omits the minimality (existence). Below we list the common patterns found in the manual diagnostic process in table 2. We also find that some of the errors in the table can be partially fixed by post-processing.

## 5.3 Effectiveness and discussion

For an intuitive comparison of the effectiveness of our active learning pipeline, we derive the CPN and NPN for three models: The first-round model which is used for the initial filtering, the final-round model generating our dataset, and the final model further fine-tuned on our dataset Lean Workbook. The results are shown in Table 3. We also listed the accuracy of MiniF2F valid and test set when an

Table 1: Accuracy by tags in Lean Workbook. These are tags that show up more than 100 times in our final dataset. The first three most common tags are sampled 10 problems for each tag, while the others are sampled 5 problems. It is worth noting that some tags are incorrect due to the mistake of the tagging model, and we will choose another sample with this tag if we consider the current one unsuitable.

| Tags | Number of samples | Sampled accuracy |
|------|-------------------|------------------|
| inequality | 46847 | 10/10 |
| algebra | 45218 | 9/10 |
| number theory | 22474 | 9/10 |
| trigonometry | 4133 | 4/5 |
| equation | 3255 | 5/5 |
| proof | 3172 | 5/5 |
| calculus | 1061 | 4/5 |
| sequence | 926 | 4/5 |
| combinatorics | 893 | 4/5 |
| series | 418 | 5/5 |
| function | 351 | 4/5 |
| modular arithmetic | 339 | 4/5 |
| induction | 285 | 5/5 |
| logarithm | 269 | 5/5 |
| limit | 224 | 3/5 |
| real analysis | 170 | 5/5 |
| Weighted Average | - | 0.935 |

Table 2: Case study for false patterns. We list the common patterns concluded during the iterative diagnostic process. This table gives one typical error for each pattern and also demonstrates one heuristic correction. Finally, the current performance column states how many portions the model can translate correctly after the iteration in our manual check.

| Pattern | Wrong example | Modified | Performance |
|---------|---------------|----------|-------------|
| Type confusion | `a,b,c : ` $\mathbb{R}$`,` `sqrt (a ^ 2 + 8 * b * c)` | `sqrt` $\to$ `Real.sqrt` | Mostly Correct |
| Continued inequalities | `a >= b >= c > 0` | `a >= b` $\wedge$ `b >= c` $\wedge$ `c > 0` | Mostly Correct |
| Missing operators | `2a+3b >= 0` | `2*a+3*b >= 0` | Mostly Correct |
| Integer division | `(a*b*c)^(1/3)` | `(a*b*c) ^((1:`$\mathbb{R}$`)/3)` | Half Correct |
| Triangle condition | $a, b, c$ are side lengths of a triangle: not translated | `(hx: a > 0`$\wedge$`b > 0`$\wedge$`c > 0)` `(hab : a + b > c)` `(hbc : b + c > a)` `(hca : a + c > b)` | Mostly Correct |
| All solutions | `(x,y)=(1,5),(2,3)` | `(x=1`$\wedge$`y=5)` $\vee$ `(x=2`$\wedge$`y=3)` | Mostly Correct |
| Solution number/sum | `(x,y)=(1,5),(2,3)` | `A : Finset {x,y|...}` `A.card=2` | Mostly Correct |
| Min/Max | The maximal of $a$ is 10: `a <= 10` | `IsGreatest {a | ...} 10` | Half Correct |
| Exist Infinite number | Unable to translate | $\forall$`N: N,`$\exists$`n > N, ....` | Mostly Correct |
| Digits | `n =abcde, a+b = ...` | `Finset {n|` `sumOflist (Nat.digits 10 n)` | Half Correct |

InternLM2-Math-Plus model is fine-tuned on MiniF2F with and without our Lean-Workbook dataset in Table 4.

Table 3: We report the CPN (compile pass number) and NPN (NLI pass number) for each model during iterations.

| Train Dataset | Model | CPN | NPN |
|---|---|---|---|
| MiniF2F + ProofNet + MultiTask | First-round Model | 136670 | 37122 |
| + Human-labeled | Final-round Model | 205079 | 57231 |
| + Lean Workbook | Final-round Model + Lean Workbook | **228928** | **82893** |

Table 4: We report the accuracy for an InternLM2-Math-Plus model fine-tuned on MiniF2F only and with our Lean-Workbook dataset.

| Train Dataset | MiniF2F-valid Acc. | MiniF2F-test Acc. |
|---|---|---|
| Mathlib | 44,3 | 37.3 |
| + Lean Workbook | **50.4** | **46.7** |

This table clearly shows the effectiveness both of our pipeline and our dataset. The human-labeled data and filtered dataset achieve a gain of over 20000 more correct samples for both the compile test and the NLI test, which promisingly indicates that this form of active learning can be further iteratively utilized. The increase in MiniF2F accuracy also demonstrates a significant improvement in performance when using our extended dataset.

The model can further enhance its pass number by adding Lean Workbook data for translation, we will also open-source this dataset (named Lean Workbook Plus). Although it shows a higher number on NLI pass rate, the human evaluation finds that this dataset makes more mistakes on the number theory problems, especially on the problem with prime numbers and maximal/minimal values.

### 5.4 Formalizing IMO problems

The accuracy table and case study table give us confidence in the performance of our model. As a high-level application, we try to translate new IMO problems using our model.

We aggregate the problems from Compfiles which have not been formalized. Each of the problems is translated 100 times under a temperature of 0.7 and we remove the wrong translations by compile test and NLI test. Finally, 23 problems with at least one correct translation passing the NLI are filtered out, and 21 problems are kept after manual evaluation, including 14 Algebra problems, 5 Number Theory problems, and 2 Combinatorics problems. We also manually checked and made slight modifications to the conclusion part if the correct answers to IMO problems are not extracted, and we ensure these translations are correct. These formal statements will be submitted to the Compfiles project. One case below shows that our model has been able to skillfully use "Finset" functions to optimize formal statements and avoid grammar mistakes. More cases are listed in Appendix B.

```
/--
IMO 1983 P5

Is it possible to choose 1983 distinct positive integers, all less than or equal to
    10^5, no three of which are consecutive terms of an arithmetic progression?
--/

theorem IMO1983_P5 :
    ∃ S : Finset ℕ, S.card = 1983 ∧ (∀ x ∈ S, x ≤ 10^5) ∧
    ∀ x ∈ S, ∀ y ∈ S, ∀ z ∈ S, x < y ∧ y < z → x + z ≠ 2 * y := by sorry
```

## 6   Conclusion

In this paper, we introduce an automatic pipeline that can translate contest-level math problems into Lean formal statements with high accuracy. Active learning proves its effectiveness in the data-sparse

scenario. We open-source Lean Workbook to help the machine learning community to improve the ability of autoformalization and automatic theorem proving.

## Limitations

We find our proposed dataset has some similar problems which is hard to apply deduplication. Furthermore, our model is focused on contest-level problems during active learning which may not be appropriate to formalize other level math problems.

## Acknowledgements

This work is supported by Shanghai Artificial Intelligence Laboratory, and funded by the project JF-P23KK00072-2-DF.

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

# A   Case study

We would give some translated examples of Lean Workbook with respected to most common tags.

## Inequality

**Natural Language problem:** For $a, b, c, d > 0, abcd = 1$ prove that $\frac{1}{1+(1+a)^2} + \frac{1}{1+(1+b)^2} + \frac{1}{1+(1+c)^2} + \frac{1}{1+(1+d)^2} \leq \frac{4}{5}$.

```
theorem lem1 (a b c d : ℝ) (hab : 0 < a) (hbc : 0 < b) (hcd : 0 < c) (hda :
    0 < d) (habc : a * b * c * d = 1) :
(1 / (1 + (1 + a) ^ 2) + 1 / (1 + (1 + b) ^ 2) + 1 / (1 + (1 + c) ^ 2) + 1 /
    (1 + (1 + d) ^ 2)) ≤ (4:ℝ) / 5 := by sorry
```

## Algebra

**Natural Language problem:** Prove that for $m \geq 5$, the sum of the factorials of the first m natural numbers is not equal to the product of the factorials of the first m odd natural numbers.

```
theorem sum_factorial_not_prod_factorial (m : ℕ) (hm : 5 ≤ m) : (Σ k in
    Finset.range m, k!) ≠ (∏ k in Finset.Icc 1 m, (2 * k - 1)!) := by sorry
```

## Number Theory & Combination

**Natural Language problem:** For every p prime number show that $p^2 \mid \binom{2p}{p} - 2$.

```
theorem p2_dvd_2pCp_2 (p : ℕ) (hp : Nat.Prime p) : p^2 | (Nat.choose (2 * p)
    p) - 2 := by sorry
```

## Equation

**Natural Language problem:** Solve in $\mathbb{R}$ : $sin(a) + 2sin(a + b + c) = 0$, $sin(b) + 3sin(a + b + c) = 0$, $sin(c) + 4sin(a + b + c) = 0$.

```
theorem sin_problem (a b c : ℝ) : sin a + 2 * sin (a + b + c) = 0 ∧ sin b +
    3 * sin (a + b + c) = 0 ∧ sin c + 4 * sin (a + b + c) = 0 ↔ ∃ (k n m :
    ℤ), a = π * k ∧ b = π * n ∧ c = π * m := by sorry
```

## Trigonometry

**Natural Language problem:** If $a, b, c$ be sidelengths of a triangle, then $\frac{a}{b+c} + \frac{b}{c+a} + \frac{c}{a+b} \leq \frac{1}{2} + \frac{3(a^3+b^3+c^3)}{(a+b+c)(a^2+b^2+c^2)}$.

```
theorem imo_1964_p1_1 (a b c : ℝ) (hx: a > 0 ∧ b > 0 ∧ c > 0) (hab : a + b
    > c) (hbc : b + c > a) (hca : a + c > b) : (a / (b + c) + b / (c + a) +
    c / (a + b) : ℝ) ≤ (1:ℝ) / 2 + (3 * (a ^ 3 + b ^ 3 + c ^ 3)) / ((a + b +
    c) * (a ^ 2 + b ^ 2 + c ^ 2)) := by sorry
```

## Proof & Complex Number

Natural Language problem: Let $a$, $b \in \mathbb{R}$ and $A = \{z \in \mathbb{C} \,|\, Re\,z = a\}$, $B = \{z \in \mathbb{C} \,|\, Im\,z = b\}$. Prove that if $M \subset \mathbb{C}$ has the properties: 1) $A \subset M$ and $B \subset M$; 2) whatever $z_1, z_2 \in M \Rightarrow z_1 + z_2 \in M$, then $M = \mathbb{C}$.

```
theorem subset_of_1d_subsets {M : Set ℂ} (a b : ℝ)    (hA : {z : ℂ | z.re =
    a} ⊆ M)    (hB : {z : ℂ | z.im = b} ⊆ M)    (hM : ∀ z1 z2 : ℂ, z1 ∈ M ∧
    z2 ∈ M → z1 + z2 ∈ M) : M = ℂ := by sorry
```

## Calculus & Limit

Natural Language problem: Prove that $\lim_{n \to \infty} \frac{1}{n} \left( \sum_{i=1}^{n} \frac{i}{\sqrt{i^2+1}} \right) = 1$

```
theorem Injective_total : ∀ ε : ℝ, ε > 0 → ∃ N : ℕ, ∀ n : ℕ, n ≥ N → |(1
    / n) * (Σ i in Finset.Icc 1 n, i / (Real.sqrt (i ^ 2 + 1))) - 1| < ε :=
    by sorry
```

## function

Natural Language problem: Find all functions $f : \mathbb{R} \to \mathbb{R}$ satisfying: for all $x, y \in \mathbb{R}$: $f(xf(x) + f(y)) = (f(x))^2 + y$

```
theorem FE (f : ℝ → ℝ):(∀ x y, f (x * f x + f y) = (f x)^2 + y) ↔ ∀ x, f
    x = x ∨ ∀ x, f x = -x := by sorry
```

## sequence & induction

Natural Language problem: Suppose that $a_n$ is a sequence such that $a_{n+1} = a_n^2 + na_n - 2$ with $a_1 = 3$, Show that $\frac{1}{a_1-2} + \frac{1}{a_2-2} + \cdots + \frac{1}{a_n-2} < 2$

```
theorem aops_1212 (n : ℕ) (a : ℕ → ℕ) (ha : a 1 = 3) (hab : ∀ n, a (n + 1)
    = (a n)^2 + n * a n - 2) : Σ k in Finset.Icc 1 n, (1 / (a k - 2)) < 2 :=
    by sorry
```

## Modular Arithmetic

Natural Language problem: Show that the cube of any integer is congruent to 0, 1, or -1 modulo 9.

```
theorem t_cubic_mod9 : ∀ t : ℤ, t^3 ≡ 0 [ZMOD 9] ∨ t^3 ≡ 1 [ZMOD 9] ∨ t^3
    ≡ -1 [ZMOD 9] := by sorry
```

## Real Analysis

**Natural Language problem:** Let $D$ be a compact subset of $\mathbb{R}$ and support that $f : D \to \mathbb{R}$ is continuous. Prove $f(D)$ is compact.

```
theorem continuous_compact_support (D : Set ℝ) (f : ℝ → ℝ)  (hD :
    IsCompact D) (hf : ContinuousOn f D) : IsCompact (Set.image f D) := by
    sorry
```

# B   IMO example

Our model also provides formalization for IMO-level problems. The translated questions focus on three types: Algebra, Number Theory, and Combinatorics.

## Algebra IMO 1975 P2

**Natural Language problem:** Let $a_1 < a_2 < a_3 < \cdots$ be positive integers. Prove that for every $i >= 1$, there are infinitely many an that can be written in the form $a_n = ra_i + sa_j$, with r, s positive integers and $j > i$.

```
theorem imo1975_p2 (a : ℕ → ℤ) (apos : ∀ i, 0 < a i) (ha : ∀ i, a i < a (i
    + 1)) (i : ℕ) : ( ∀ i n0:ℕ , ∃ n, n0 ≤ n ∧ ∃ r s : ℕ, ∃ j : ℕ, a n =
    r * a i + s * a j ∧ i < j ∧ 0 < r ∧ 0 < s ):= by sorry
```

## Algebra IMO 1977 P4

**Natural Language problem:** Define $f(x) = 1 - a\cos x - b\sin x - A\cos 2x - B\sin 2x$, where a, b, A, B are real constants. Suppose that $f(x) \geq 0$ for all real x. Prove that $a^2 + b^2 \leq 2$ and $A^2 + B^2 \leq 1$.

```
theorem imo1977_p4 (f : ℝ → ℝ) (a b A B : ℝ)  (h₀ : ∀ x, f x = 1 - a *
    Real.cos x - b * Real.sin x - A * Real.cos (2 * x) - B * Real.sin (2 *
    x))  (h₁ : ∀ x, f x ≥ 0) : a ^ 2 + b ^ 2 ≤ 2 ∧ A ^ 2 + B ^ 2 ≤ 1 := by
    sorry
```

## Number Theory IMO 1978 P1

**Natural Language problem:** m and n are positive integers with $m < n$. The last three decimal digits of $1978^m$ are the same as the last three decimal digits of $1978^n$. Find m and n such that m + n has the least possible value.

```
theorem imo1978_p1 (m n : ℕ) (hmn: m < n) (hmn2: m = 3 ∧ n=103) : (1978^m)
    % 1000 = (1978^n) % 1000) ∧ (∀ m' n' : ℕ, m' < n' ∧ (1978^m') % 1000 =
    (1978^n') % 1000 → m + n ≤ m' + n') := by sorry
```

---

**Number Theory IMO 1982 P4**

**Natural Language problem:** Prove that if $n$ is a positive integer such that the equation $x^3 - 3xy^2 + y^3 = n$ has a solution in integers x, y, then it has at least three such solutions. Show that the equation has no solutions in integers for $n = 2891$.

```
theorem imo1982_p4 (n : ℕ) (hn : 0 < n) (hxy : ∃ x y : ℤ, x^3 - 3 * x * y^2 +
    y^3 = n) : (n ≠ 2891) ∧ ∃ x1 x2 x3 y1 y2 y3 : ℤ, (x1^3 - 3 * x1 * y1^2
    + y1^3 = n ∧ x2^3 - 3 * x2 * y2^2 + y2^3 = n ∧ x3^3 - 3 * x3 * y3^2 +
    y3^3 = n ∧ (x1 ≠ x2 ∨ y1 ≠ y2) ∧ (x1 ≠ x3 ∨ y1 ≠ y3)  ∧ (x2 ≠ x3 ∨
    y2 ≠ y3)) := by sorry
```

---

**Combinatorics IMO 1978 P6**

**Natural Language problem:** An international society has its members from six different countries. The list of members has 1978 names, numbered $1, 2, \ldots, 1978$. Prove that there is at least one member whose number is the sum of the numbers of two (not necessarily distinct) members from his own country.

```
theorem imo1978_p6 (n : ℕ) (hn : n = 1978) (C : Fin n → Fin 6) : ∃ i : Fin
    n,  ∃ j : Fin n, ∃ k : Fin n,  C i = C j ∧ C j = C k ∧ i ≠ k ∧ (i:ℕ
    ) + (k:ℕ ) = (j:ℕ ) + 1 := by sorry
```

---

## C  Dataset card

1. Our dataset contains 57231 problems in the split of Lean Workbook and 82893 problems in the split of Lean Workbook Plus. We provide the natural language statement, answer, formal statement, and formal proof (if available) for each problem. These data can support autoformalization model training and searching for proofs.

2. We open-source our code at `https://github.com/InternLM/InternLM-Math` and our data at `https://huggingface.co/datasets/InternLM/Lean-Workbook`.

3. Croissant metadata URL: `https://huggingface.co/api/datasets/internlm/Lean-Workbook/croissant`.

4. The license of our dataset is Apache 2.0.

5. We will host our dataset in Huggingface and our code in GitHub. We will maintain this dataset with further improvement.

6. DOI of dataset: 10.57967/hf/2399

