# OpenReview forum: "Lean Workbook: A large-scale Lean problem set formalized from natural language math problems"
_NeurIPS.cc/2024/Datasets_and_Benchmarks_Track — NeurIPS 2024 Track Datasets and Benchmarks Poster_

### Official Review · Reviewer_eHjp · 2024-07-25
**Iterative autoformalization pipeline that renders a large-scale Lean problem set**

**Rating:** 8
**Confidence:** 5
**Correctness:** Claims in the paper seem correct.

**Review:**

I have broken down my review into the next few specific sections. Generally, I like this paper as elaborated under the "Strengths" section. I list three questions under the "Opportunities For Improvement" section, one under "Limitations", and some minor suggestions in "Clarity". Should the authors have space, I would encourage them to elaborate a bit more on the related works, as I discuss in the "Relation To Prior Work" section. Overall I believe this paper is a good candidate for acceptance into NeurIPS, and look forward to hearing the authors' responses to the questions.

**Strengths:**

I like how this paper tackles one of the most important issue in the field -- learning in data-sparse scenarios. The workflow is natural and complete, and the performance is verified to be great. I especially appreciate the authors' effort in seriously evaluating their results. The curated datasets can be very valuable resources for the field of AI for mathematics. It is also very relevant to the broader AI community, since theorem proving is one of the most difficult open problems in AI.

**Additional Feedback:**

None. All feedback is already listed above.

**Clarity:**

The paper is very clearly written. The flow is very natural and the detail level is just right. Two minor (picky) comments:

1. In Sec. 2, the authors write "The Lean language can claim a theorem and prove it by tactics or pretend to complete the proof using 'sorry'." To be a bit picky about the concepts, Lean itself does not autonomously do conjecture or prove things; also, although tactics are the main strategy for theorem proving in Lean, term-based proofs exists too. Thus it's probably better to reword this sentence as "In Lean, users can define a theorem and prove it by tactics or terms ...".

2. It's probably better to have consistent names for artifacts. E.g. translation model v.s. translate model. Both names appear during the writing.

**Documentation:**

Yes, the documentation seems clear.

**Ethics:**

There seems to be no ethical concern.

**Limitations:**

Limitations have been comprehensively discussed. There seems to be no direct societal impact with this work.

By the way, I notice that the authors report that "our proposed dataset has some similar problems which is hard to apply deduplication". I wonder how severe is the duplication problem? It's best to have some quantitative sense by e.g. some similarity measures, otherwise I would at least be interested in knowing some evidence for an estimated qualitative severity.

**Opportunities For Improvement:**

1. In Sec. 4.1, the authors write "Concretely, we append "Show that it is {answer}." to the original natural questions, while the proof goal in formal statements is changed to prove the solved answer should be the gold one." This brute-force strategy of turning a question-answer pair into a theorem has been under debate for quite some time, as it can render the formalized theorem to be of a lower difficulty. Are the authors aware of this limitation?

2. In Sec. 4.1, when checking whether the math problems are well-defined or not, the prompt explicitly instructs "be sure not to fail well-defined questions" in the end. Does this mean the authors would like to minimize Type-1 errors during this stage? If so, this would satisfy Type-2 errors. Is this choice because Type-2 errors were observed to occur unfrequently during experiments? Why that is / is not the case?

3. In the last paragraph of Sec. 4.1, the authors write "More than three-thirds of the samples are labeled with algebra-relevant tags, while geometry-related tags are rarely witnessed." Three-thirds = 3/3 = 1, that seems incorrect?

**Relation To Prior Work:**

The discussion of prior work is brief yet clear. Since this work reports progress on datasets, it can be beneficial to include another subsection of related works on general dataset curation / in-environment LM evaluation in the field.

E.g. Dataset curation:

[1] **LeanDojo** (classic dataset in Lean): Kaiyu Yang, Aidan Swope, Alex Gu, Rahul Chalamala, Peiyang Song, Shixing Yu, Saad Godil, Ryan J Prenger, Anima Anandkumar. LeanDojo: Theorem Proving with Retrieval-Augmented Language Models. In Neural Information Processing Systems (NeurIPS), 2023.

[2] **CoqGym** (classic dataset in Coq): Kaiyu Yang, Jia Deng. Learning to Prove Theorems via Interacting with Proof Assistants. In Proceedings of the 36th International Conference on Machine Learning, PMLR 97:6984-6994, 2019.

[3] **miniCodeProps** (new dataset in Lean): Evan Lohn, Sean Welleck. miniCodeProps: a Minimal Benchmark for Proving Code Properties. *arXiv preprint arXiv:2406.11915*, 2024.

[4] **Lean Copilot** (native LM eval framework): Peiyang Song, Kaiyu Yang, Anima Anandkumar. Towards Large Language Models as Copilots for Theorem Proving in Lean. *arXiv preprint arXiv:2404.12534*, 2024.

[5] **LLMStep** (server-based LM eval framework): Sean Welleck, Rahul Saha. LLMSTEP: LLM proofstep suggestions in Lean. *arXiv preprint arXiv:2310.18457*, 2023.

... and more. The authors have actually already cited some of such dataset/evaluation works in the field that carry important values yet not necessarily restricted to autoformalization abilities, but it may help to have a short subsection to discuss them together in the paper.

Also, this work addresses a similar task as the MMA paper (which the authors did cite). Since both works generate formal-informal pairs, it would be great if some comparisons or at least specific discussions can be performed.

**Summary And Contributions:**

This work introduces an autonomous pipeline to translate contest-level math problems into formal Lean code. High accuracy is achieved and the quality of generation is verified in three stages (one of them is human expert evaluation). This works well in under data sparsity, as proven via active learning. The curated set of formal-informal theorem pairs has a large size of about 57K; combined with a theorem proving LLM, some of the newly formalized theorems can be proved, which additionally offer data for NTP task.

---

> ### Author Rebuttal · Authors · 2024-08-13
>
> Thank you for raising these points regarding potential problems in our work. Please allow me to explain them one by one.
>
> Review 1:  In Sec. 4.1, the authors write "Concretely, we append "Show that it is {answer}." to the original natural questions, while the proof goal in formal statements is changed to prove the solved answer should be the gold one." This brute-force strategy of turning a question-answer pair into a theorem has been under debate for quite some time, as it can render the formalized theorem to be of a lower difficulty. Are the authors aware of this limitation?
>
> Response:
> Actually, we are working on supporting translating problems into Compfiles-style which could leave the true answer as sorry and avoid such transformation. We will try to improve the performance of this, and will try to update this in future versions of Lean-workbook.
>
>
> Review 2: In Sec. 4.1, when checking whether the math problems are well-defined or not, the prompt explicitly instructs "be sure not to fail well-defined questions" in the end. Does this mean the authors would like to minimize Type-1 errors during this stage? If so, this would satisfy Type-2 errors. Is this choice because Type-2 errors were observed to occur unfrequently during experiments? Why that is / is not the case?
>
> Response:
> Regarding the prompt in Section 4.1, you are correct that the instruction "be sure not to fail well-defined questions" was added to minimize Type-1 errors. The original prompt did not include this instruction, but after reviewing the results, we observed that the majority of errors were indeed Type-1. We believe that this is largely due to most ill-defined problems arising from extraction errors, leading to semantically incoherent or incomplete statements and is easy for the model to discriminate. Therefore, we adjusted the prompt to address this issue and improve the accuracy of the evaluation.
>
>
> Review 3: In the last paragraph of Sec. 4.1, the authors write "More than three-thirds of the samples are labeled with algebra-relevant tags, while geometry-related tags are rarely witnessed." Three-thirds = 3/3 = 1, that seems incorrect?
>
> Response:
> We apologize for the mistake, and we have corrected it to accurately reflect the distribution of algebra-relevant (three-fourths) tags.
>
>
> Review 4: By the way, I notice that the authors report that "our proposed dataset has some similar problems which is hard to apply deduplication". I wonder how severe is the duplication problem? It's best to have some quantitative sense by e.g. some similarity measures, otherwise I would at least be interested in knowing some evidence for an estimated qualitative severity.
>
> Response:
> Our original problems were sourced from AoPS, and it indeed includes samples that are essentially the same problem but expressed differently. These cases are challenging to deduplicate due to their varied formulations. As this dataset is intended for training purposes, we have acknowledged the existence of this issue but have not conducted a detailed exploration of the impact that these relatively few duplicated samples may have on the overall dataset quality. We recognize the importance of addressing this in future work and will consider implementing similarity measures to quantify and analyze the severity of duplication.

---

> > ### Comment · Reviewer_eHjp · 2024-08-13
> > **Reviewer Response to Author Rebuttal**
> >
> > Thank the authors for the quick rebuttal.
> >
> > * Re R1, R4: Sounds good, look forward to your future versions that improve on these aspects.
> > * Re R2: This makes sense. I think this is a pretty interesting observation itself.
> > * Re R3: Thanks for correcting the typo.
> >
> > By the way, I saw you attached a new version of the paper but you didn't mention anything specific in it during your rebuttal. Would you like me to consider any part in the new version during our discussion? If yes, could you briefly summarize what you have changed and what you would like me to look at?

---

> > > ### Comment · Reviewer_eHjp · 2024-08-13
> > > **Any responses to my other comments?**
> > >
> > > In addition, do the authors have any responses to my comments under the `Clarity` and `Relation to Prior Work` sections? No worries if not.

---

> > > > ### Author Rebuttal · Authors · 2024-08-21
> > > >
> > > > Thank you for your comments regarding Clarity and Relation to Prior Work. We have revised the description in Section 2 and standardized the terminology by consistently referring to the "translation model".  Additionally, we have added a "Data Curation" subsection in Section 3, incorporating citations of the papers you recommended to further enrich our content.
> > > >
> > > > The PDF below (and in the later version) is the revised paper, which includes changes based on your feedback as well as the comments from other reviewers, summarized as follows:
> > > > 1. Corrected three-thirds to three-fourths in Sec 4.1
> > > > 2. Standardized the terminology to the "translation model".
> > > > 3. Added a "Data Curation" subsection in Section 3
> > > > 4. Added Table 4 in Section 5.3 to validate the effectiveness of our dataset: we conducted experiments using the 7B InternLM-Math-Plus model, fine-tuned with our proposed datasets, including both Lean Workbook and Mathlib data. The results were as follows: MiniF2F-Test Accuracy: 46.7% / MiniF2F-Validation Accuracy: 50.4% In comparison, when the model was fine-tuned solely with Mathlib data, the results were: MiniF2F-Test Accuracy: 37.3% / MiniF2F-Validation Accuracy: 44.3%
> > > > 5. Added details on quality control in Section 4.2, line 174: Our evaluators contain three human experts who are familiar with both Lean and contest-level math problems and have won a prize in the National Mathematical Olympiad Contest. Each evaluator was assigned an equal number of problems, ensuring a balanced distribution of the workload. On average, each problem required approximately two to five minutes for evaluation.

---

> > > > > ### Comment · Reviewer_eHjp · 2024-08-21
> > > > > **Changes look great; raising score**
> > > > >
> > > > > These changes look great, thank the authors for the efforts! I think it's pretty evident even from my initial reviews that I have quite liked this work. As I still believe in the strengths I pointed out, all the questions I raised have been addressed during the discussion period too. I am now further raising my score to recommend a clear accept, and hope the authors keep up with the solid work in this important field!

---

### Official Review · Reviewer_cgHc · 2024-07-25
**Review 1169**

**Rating:** 6
**Confidence:** 4
**Correctness:** The dataset is constructed in a sound…
**Clarity:** The paper is well-written and easy to…

**Review:**

The paper proposed a sound and effective method to auto-formalize contest-level word problems. The released dataset samples are hard and mostly correct. However, the authors do not show the potential of these samples by inspecting whether they can improve current LLMs.

**Strengths:**

- The proposed active-learning framework is sound and effective. It applies several verifies to filter the incorrect translations efficiently and incorporate human experts to ensure correctness. The human assessment shows that 93.5% of the samples are correct, verifying the correctness of the constructed dataset.

- The math problems in the proposed dataset are harder than most previous datasets. These different and harder samples can expand training distribution and are very likely to improve current LLMs.

**Additional Feedback:**

The authors can include more experiments to show the data can help LLMs in solving math problems.

**Documentation:**

The paper gives details of the data collection and organization.

**Ethics:**

The paper has no ethical concerns.

**Limitations:**

The authors adequately addressed the limitations

**Opportunities For Improvement:**

- The paper does not verify the effectiveness of the dataset sample. For example, whether a model finetuned with these extra data can improve the performance on miniF2F.

- The proposed datasets only contain problem-statement pairs. It does not have sufficient formal solutions to support auto-formalizing the solving process.

**Relation To Prior Work:**

The paper includes sufficient related work and discusses the difference.

**Summary And Contributions:**

The paper proposes a new auto-formalization method to translate math problems into formal statements in Lean. It iteratively finetunes an LLM through translating problems, identifying the incorrect translation, and labeling them with human experts. Based on the proposed method, the paper constructs a new dataset consisting of contest-level math problems(including IMO) and their statement.

---

> ### Author Rebuttal · Authors · 2024-08-13
>
> Review1: The paper does not verify the effectiveness of the dataset sample. For example, whether a model finetuned with these extra data can improve the performance on miniF2F.
>
> Response:
> We appreciate the reviewer's concern regarding the effectiveness of the dataset samples. To address this, we conducted experiments using the 7B InternLM-Math-Plus model, fine-tuned with our proposed datasets, including both Lean Workbook and Mathlib data. The results were as follows:
> MiniF2F-Test Accuracy: 46.7% / MiniF2F-Validation Accuracy: 50.4%
> In comparison, when the model was fine-tuned solely with Mathlib data, the results were:
> MiniF2F-Test Accuracy: 37.3% / MiniF2F-Validation Accuracy: 44.3%
> which demonstrates a significant improvement in performance when using our extended dataset, validating its effectiveness. This result has been added to the paper in Table 4 in Section 5.3. Also, after incorporating the extra data, we observed an improvement in both the compilation and NLI (Natural Language Inference) rates.
>
>
> Review 2: The proposed datasets only contain problem-statement pairs. It does not have sufficient formal solutions to support auto-formalizing the solving process.
>
> Response:
> We agree that solving these problems is a complex task, and providing a comprehensive dataset with complete solutions is beyond the scope of this work. One of the primary goals of releasing this dataset is to facilitate future research that could focus on expert iteration on the Lean Workbook, aiming to enhance the performance of automated theorem proving (ATP). We believe that our dataset provides a valuable foundation for this endeavor and hope it will inspire further advancements in this area.

---

### Official Review · Reviewer_uCYe · 2024-08-01
**Review for the paper**

**Rating:** 7
**Confidence:** 3
**Correctness:** yes, just some details are missing, s…
**Clarity:** yes

**Review:**

The paper make a good contribution on alleviating the problem of limited formalized math problems. The pipeline is simple. Even with human-in-the-loop, the method should be much easier to scale up than only human efforts.

The con is mainly the details on quality control are not clear. The paper mention manual inspection / evaluation but didn't elaborate on who are the evaluator, and how many evaluator evaluate each problem, etc... And the number of problems for evaluating accuracy is quite small only 5 to 10 problems per type, the paper should have more problems evaluated like 50 problems per type. Then some detailed error analysis is also helpful to understand what are the weakness of the translation model and provide insights for future work.

**Strengths:**

The pipeline is simple and can be scale up.
The community will benefit from the data and the translation model.

**Additional Feedback:**

see above

**Documentation:**

yes

**Limitations:**

Copy from the review above:

The con is mainly the details on quality control are not clear. The paper mention manual inspection / evaluation but didn't elaborate on who are the evaluator, and how many evaluator evaluate each problem, etc... And the number of problems for evaluating accuracy is quite small only 5 to 10 problems per type, the paper should have more problems evaluated like 50 problems per type. Then some detailed error analysis is also helpful to understand what are the weakness of the translation model and provide insights for future work.

**Opportunities For Improvement:**

Details on annotator/evaluator are missing, as well as small number of problems for quality evaluation.

**Relation To Prior Work:**

yes

**Summary And Contributions:**

The paper focus on formalizing natural language math problems to formal language such as Lean as well as back-translation. For this, the paper introduces a active learning human-in-the-loop pipeline that involves filtering and iterative training the theorem translation model. In the end, the paper produces a synthetic dataset that contains 57k formalized math problems, and 5K of them have solution generated by a math proving model.

---

> ### Author Rebuttal · Authors · 2024-08-13
>
> We appreciate the reviewer’s feedback regarding the quality control details and the number of problems used for evaluating accuracy. We would like to clarify the following points:
> Our evaluators contain three human experts who are familiar with both Lean and contest-level math problems and have won a prize in the National Mathematical Olympiad Contest. Each evaluator was assigned an equal number of problems, ensuring a balanced distribution of the workload. On average, each problem required approximately two to five minutes for evaluation. This has been added to the paper in Section 4.2, line 165. Due to the availability of evaluators, we were unable to provide a more extensive analysis with a larger sample size, as each round of training would require the same number of problems to be re-evaluated, significantly increasing the overall evaluation effort. However, we believe that the error analysis presented in the paper covers a broad spectrum of the model's performance, providing valuable insights into its strengths and weaknesses. We agree that expanding the evaluation to include more problems and a more detailed quality control process would further enhance the robustness of our findings, and we plan to address these aspects in our future work.

---

> > ### Comment · Reviewer_uCYe · 2024-08-29
> >
> > Thanks for your rebuttal, I raise my score to 7.

---

### Decision · Program_Chairs · 2024-09-26

**Decision:**

Accept (Poster)

**Comment:**

This paper introduces an approach to translating the math problem into a formal Lean code. The main idea is to use a human-in-the-loop framework that iteratively trains an LLM on translation problems, filters incorrect samples and relabels them by human. The resulting synthetic dataset consists of 57k samples for math problems.


Overall, the proposed approach is straightforward and has good scalability property. Furthermore, the curated datasets could be valuable resources for AI for mathematics and reasoning. The concerns raised by reviewers were properly addressed.

To sum up, I’d like to recommend it acceptance.